# A Putative NADPH Oxidase Gene in Unicellular Pathogenic *Candida glabrata* Is Required for Fungal ROS Production and Oxidative Stress Response

**DOI:** 10.3390/jof10010016

**Published:** 2023-12-27

**Authors:** Maoyi Lin, Yao Huang, Kanami Orihara, Hiroji Chibana, Susumu Kajiwara, Xinyue Chen

**Affiliations:** 1School of Life Science and Technology, Tokyo Institute of Technology, Yokohama 226-8501, Japan; lynnmaoyi@foxmail.com (M.L.); huangyao0727@gmail.com (Y.H.); orihara.k.ab@m.titech.ac.jp (K.O.); kajiwara.s.aa@m.titech.ac.jp (S.K.); 2Medical Mycology Research Center, Chiba University, Chiba 263-8522, Japan; chibana@faculty.chiba-u.jp

**Keywords:** *Candida glabrata*, NADPH oxidase, ROS production, oxidative stress response

## Abstract

Most previous studies on fungal NADPH oxidases (Nox) focused on multicellular fungi and highlighted the important roles of Nox-derived reactive oxygen species (ROS) in cellular differentiation and signaling communication. However, there are few reports about Nox in unicellular fungi. A novel *NOX* ortholog, *CAGL0K05863g* (named *CgNOX1*), in *Candida glabrata* was investigated in this study. Deletion of *CgNOX1* led to a decrease in both intracellular and extracellular ROS production. In addition, the *Cgnox1∆* mutant exhibited hypersensitivity to hydrogen peroxide and menadione. Also, the wild-type strain showed higher levels of both *CgNOX1* mRNA expression and ROS production under oxidative stress. Moreover, the absence of *CgNOX1* resulted in impaired ferric reductase activity. Although there was no effect on in vitro biofilm formation, the *CgNOX1* mutant did not produce hepatic apoptosis, which might be mediated by fungal Nox-derived ROS during co-incubation. Together, these results indicated that the novel *NOX* gene plays important roles in unicellular pathogenic *C. glabrata* and its interaction with host cells.

## 1. Introduction

*Candida* species are major fungal pathogens and cause opportunistic *Candida* infections, the incidence of which has significantly increased during the past decades [1,2,3]. *Candida* infections are a commonly recognized complication in clinical cases, as the fungi can invade the gastrointestinal tract and reach the host organs, especially in immunocompromised individuals [4]. Among non-*Candida albicans* species (NCAC) species, *C. glabrata* is known as one of the most significant opportunistic fungi for humans, ranking second to fourth depending on the geographic region [5]. Different from the diploid *C. albicans,* which can switch from yeast to hyphal growth, *C. glabrata* was identified as haploid cells and grows as a unicellular form under normal conditions without morphological change. It shows a closer phylogenetic relationship with the budding yeast *Saccharomyces cerevisiae* than *C. albicans* and other NCAC species.

Reactive oxygen species (ROS) are small oxygen-derived molecules that are generated as by-products of biochemical reactions. In addition, the common processes of respiration and photosynthesis, NADPH oxidase (a kind of Nox) is another important source to catalyze the production of ROS from oxygen and NADPH. The Nox superfamilies in mammal cells consist of seven members, Nox1-5 and Duox1-2, with various tissue distributions and activation mechanisms [6]. Their ‘oxidative burst’ roles were proposed in host defense responses to pathogen infections as well as in signaling pathways for cell proliferation and differentiation during tissue repair. *NOX* deficiency might result in immunosuppression, while excess ROS production leads to cellular stress.

In the fungal kingdom, Nox enzymes have been studied over the past two decades, focusing mainly on multicellular filamentous fungi. These studies have revealed three different Nox subfamilies, NoxA, NoxB, and NoxC, highlighting the crucial roles played by Nox-derived ROS in various physiological processes and cellular differentiation, including sexual fruiting body development, ascospore germination, and hyphal formation [7]. For example, deletion of the NoxA homologue *NOX1* in *Podospora anserina* impairs sexual fruiting body differentiation, the growth of aerial hyphae, and pigmentation [8].

However, it was proposed that the presence of *NOX* is only associated with multicellularity [9]. Additionally, there were few reports about Nox enzymes in unicellular fungi until the discovery of a Nox ortholog in the non-pathogenic yeast *S. cerevisiae* [10]. This Nox, known as Yno1p (Aim14p), was reported to be localized in the endoplasmic reticulum and required for extramitochondrial ROS generation in this yeast. Among *Candida* species, a recent paper focused on Fre8p, a *NOX* family member, in *C. albicans* and proposed its role in the production of ROS bursts during the morphological transition from the unicellular yeast form to the multicellular hyphal one [11]. The lack of *FRE8* resulted in a deficiency of hyphal development and biofilm formation during invasion in a mouse model, but not in vitro biofilm formation. In addition, it was reported that Fre8p was necessary to induce apoptosis of human cells by *C. albicans* [12].

In this work, we investigated a putative *NOX* gene in the unicellular pathogen *C. glabrata*. This study represents the first investigation of a *NOX* ortholog, CAGL0K05863g (named *CgNOX1*), in *C. glabrata*. Our findings provide evidence that this gene is involved in fungal ROS generation, oxidative stress responses, and ferric reductase activity. However, it does not appear to have a significant impact on in vitro biofilm formation. Furthermore, our results suggest that *CgNOX1* plays an important role in ROS induction during co-incubation with human hepatocytes. This study provides novel insights into the roles of NOX enzymes in *C. glabrata*.

## 2. Materials and Methods

### 2.1. Fungal Strains and Cultural Conditions

The strains used in this study are listed in Table 1. *C. glabrata* strains were grown in YPD medium (1% yeast extract, 2% peptone, 2% glucose) or SC (0.67% Bacto Yeast Nitrogen base without amino acids, 0.079% complete supplement mixture (CSM, Funakoshi, Tokyo, Japan), and 2% glucose at 30 °C. Solid media were supplemented with 2% agar.

### 2.2. Plasmid Construction and Transformation of C. glabrata

The *C. glabrata* mutants were obtained from a *C. glabrata* mutant library and constructed using the method described previously [13]. Briefly, the target gene was deleted by a DNA replacement cassette including the *CgHIS3* gene through homologous recombination in the parental strain KUE100. The recombination locus was verified by PCR using the following primer pairs: pHIScheckF and pHIScheckR.

The construction of the reintegrated strain was similar to the previous method [13]. The *CgNOX1* ORF with a 500 bp promoter and a 200 bp terminator was amplified from the template of *C. glabrata* wild-type (CBS138 strain) genomic DNA using the primers pNox1compF and pNox1compR. The total *CgNOX1* (2.5 kb) amplified fragment and the vector pZeoi_comp606 were ligated to construct pZeoi_NOX1. The primers pChr606F1 and pChr606R1 were used to amplify the cassette from pZeoi_NOX1, and the purified product was transformed into the non-coding region on chromosome F position 605,901–606,015 of the *nox1Δ* mutant by the usual lithium acetate method [13]. The primers pZeoORFcheckF1 and pchr606check were used to confirm the successful transformation at the right position. All primers used for plasmid construction are listed in Table 2.

### 2.3. Measurement of Fungal Intercellular and Extracellular ROS Production

*C. glabrata* strains were grown in SC medium at 30 °C with shaking overnight and cultured in fresh medium until the early exponential phase. For hydrogen peroxide (H_2_O_2_, Fujifilm, Osaka, Japan)-treated cells, cells were exposed to 10 mM H_2_O_2_ in SC medium for 60 min at 30 °C. In the untreated condition, the H_2_O_2_ was replaced with SC medium. Then, the cells were washed and stained with 10 μM 2′, 7′-dichlorodihydro-fluorescein deacetate (CM-H2DCFDA, Invitrogen, Waltham, MA, USA) for 30 min at 37 °C and immediately observed for fluorescence signals using an LSM 700 laser scanning confocal microscope (Carl Zeiss, Jena, Germany). The Image J software (1.53t version) is used to present relative semi-quantification data for ROS levels, in which Ostu is set as the threshold and the area mean fluorescent intensity data were calculated for at least 50 individual cells. The fungal ROS production was also measured by a chemiluminescence assay with lucigenin (N,N′-dimethyl-9-9-biacridinium dinitrate; Adipogen, San Diego, CA, USA) [15]. The early exponential cells were seeded in 96-well microplates with 100 μL per well and mixed with 100 μL of 0.1 mM lucigenin in phosphate-buffered saline (PBS: 137 mM NaCl, 2.7 mM KCl, 10 mM Na_2_HPO_4_, and 1.8 mM KH_2_PO_4_) buffer. The luminescence was measured at 37 °C using a microtiter plate reader (Varioskan LUX, Thermo Fisher Scientific, Rockford, IL, USA).

### 2.4. Spot Assay to Test Sensitivity to Oxidizing Agents

*C. glabrata* strains were grown in SC medium overnight at 30 °C and then diluted to 10^7^ cells/mL. Serial 10-fold dilutions of cells were prepared, and 3 μL of each serial dilution was spotted onto SC agar plates containing different oxidizing agents. The final concentrations of oxidizing agents were 10 mM H_2_O_2_, 1 mM diamide (Fujifilm), 0.4 mM cumene hydroperoxide (CHP; TCI Chemicals, Tokyo, Japan), and 0.05 mM menadione (Sigma-Aldrich, St. Louis, MO, USA). The plates were incubated at 30 °C for 1–3 days.

### 2.5. 2,3,5-Triphenyltetrazolium Chloride (TTC) Reduction Overlay Assay

The overlay assay with TTC (Sigma-Aldrich) was performed based on the previous published paper with some modifications [16]. In brief, overnight SC-grown *C. glabrata* strains were subcultured to obtain cells in the early-exponential growth phase. The cells were collected and diluted to 10^7^ cells/mL. Each dilution was spotted onto SC-agar plates (5 μL/spot) and incubated at 30 °C for 1 day. Then, the colonies were overlaid with agarose (1.5% in TAE buffer; TAE: 40 mM Tris, 20 mM acetic acid, and 1 M EDTA disodium salt) containing 0.01% TTC. The plates were incubated at 30 °C and photographed at the indicated time points. The reduction of TTC resulted in the formation of a dark red, insoluble formazan compound at the cell surface.

### 2.6. Extracellular Ferric Reductase Assay

Ferric reduction activity was measured based on the previously published method with a slight modification [17]. *C. glabrata* strains were grown in SC medium at 30 °C with shaking overnight. The cells were transferred to fresh SC medium with 100 μM of the iron chelator bathophenanthroline disulfonic acid (BPS, Wako, Osaka, Japan) until reaching the mid-exponential phase. After washing with ice-cold distilled water, the cells were resuspended in ice-cold assay buffer (5% glucose, 50 mM sodium citrate, pH 6.5) for 10 min of incubation at 30 °C. The cell suspension was then added with the same volume of reductase buffer (5% glucose, 50 mM sodium citrate, pH 6.5, 1 mM FeCl_3_, 1 mM BPS) for a further 5 min of incubation. The absorbance was read at 520 nm, and 1 mM dithiothreitol (DTT; Nacalai Tesque, Kyoto, Japan) was used as the positive reductant control.

### 2.7. XTT Assay and SEM Observation of Biofilms

Biofilms were formed according to an in vitro method, and the metabolic activity was measured as described previously [12]. Briefly, overnight culture cells were diluted to 1 × 10^7^ cells/mL in RPMI-1640 medium (Nacalai Tesque) and incubated in 96-well flat-bottom polystyrene plates (IWAKI, Tokyo, Japan) for 1.5 h at 37 °C for adhesion. The cells were then washed with PBS buffer and incubated with 100 μL of fresh RPMI medium for another 24 h. To measure the metabolic activity of biofilms, a 100 μL aliquot of 2,3-Bis (2-methoxy-4-nitro-5-sulfophenyl)-5-[(phenylamino) carbonyl]-2H-tetrazolium-5 carboxanilide (XTT; Cayman Chemical, Ann Arbor, MI, USA) with menadione was added into each well and then incubated for 2 h at 37 °C. The colorimetric change was measured at 490 nm using a microtiter plate reader (VarioskanLUX; Thermo Fisher Scientific). To observe the biofilm microstructure, biofilms were formed on 3 mm cubic silicone sponges treated with 2% glutaraldehyde and 1% osmium tetroxide, as described previously [18]. A desktop scanning electron microscope, PhenomTM-Pro-X (Phenom-World, Eindhove, The Netherlands), was used for biofilm observation.

### 2.8. Measurement of Hepatocyte (HC) Cellular Transglutaminase (TG) Activity and ROS Yield during Fungal Co-Incubation

For the co-incubation assay, the fungi were pre-cultured in liquid YPD medium with shaking at 180 rpm for 16 h at 30 °C. The pre-cultures were washed twice with PBS and adjusted to a concentration of 2 × 10^7^ cells/well in 6-well plates. Human HC cells (Cell Systems, Kirkland, WA, USA) were grown in CSC complete medium (Cell Systems) at 37 °C in a humidified incubator with 5% CO_2_. For co-incubation with the fungus, cells were adjusted to a concentration of 2 × 10^5^ cells/well in 6-well plates with round coverslips in Dulbecco’s modified Eagle’s medium (DMEM; Thermo Fisher Scientific) with 10% heated-inactivated fetal bovine serum (FBS). After overnight culture, the cells were incubated for 4 h in 3 mL of phenol red free DMEM with 0.2% FBS and co-incubated with 2 × 10^7^ cells/well fungi in the presence of 0.2 mM 5-(biotinamido) pentylamine (5-BAPA; Thermo Fisher Scientific) and 0.1 mM aminoguanidine for 24 h. After the co-incubation period, cells were treated as mentioned previously [12]. Briefly, cells were fixed with a 10% formaldehyde solution and stained with streptavidin-tetramethylrhodamine isothiocyanate (TRITC; Jackson ImmunoResearch Laboratories, West Grove, PA, USA) and 4′,6-diamidino-2-phenylindole (DAPI; Nacalai Tesque) dye. Both the blue fluorescence signals from DAPI-stained nuclei and the green fluorescence signals from TRITC staining 5-BAPA, representing TG activity, were detected with a LSM 780 laser scanning confocal microscope (Carl Zeiss).

For ROS detection, HC cells and *C. glabrata* were co-incubated as described above in the phenol red free DMEM medium for 24 h and treated with 5 μM CM-H2DCFDA for 30 min at 37 °C. In some cases, the fungal cells were treated with 10 mM N-acetyl cysteine (NAC; Sigma-Aldrich) in DMEM before co-incubation. The amount of NAC used was evaluated as non-cytotoxic and could effectively inhibit ROS.

### 2.9. RNA Extraction and Quantitative Real-Time PCR (qRT-PCR) Analysis

For the oxidative stress-treated samples, *C. glabrata* was treated with 5 mM H_2_O_2_ or 0.05 mM menadione in SC medium at 30 °C for 1 h. For the co-incubation samples, insert cups with a 0.4 µm pore size membrane (Corning, NY, USA) were used to separate hepatocytes and *C. glabrata*. For the single-incubation samples, the conditions were identical, except for the absence of hepatocytes. After culturing at the indicated time points, fungal cells were harvested and washed with DEPC-treated water. Total RNA was extracted using the hot phenol method [19]. The total RNA samples were reverse transcribed to first-strand cDNA using the ReverTra Ace qPCR RT Master Mix with gDNA Remover kit (Toyobo, Osaka, Japan). Real-time PCR was performed using THUNDERBIRD SYBR qPCR Mix (Toyobo) under the following conditions: 1 cycle at 95 °C for 1 min, followed by 40 cycles of 95 °C for 15 s, 60 °C for 30 s, and 72 °C for 1 min. All qRT-PCR primers are listed in Appendix A. *ACT1* mRNA was used for normalization, and the relative expression of genes was calculated by the 2^−ΔΔCt^ method [20].

### 2.10. Statistical Analysis

Quantitative data are shown as the means ± standard deviation of three independent experiments. A *p* value < 0.05 was considered statistically significant, as determined by a two-tailed unpaired *t*-test or one-way ANOVA.

## 3. Results

### 3.1. Identification of the Putative C. glabrata NADPH Oxidase Family Gene

To identify the NADPH oxidase gene family of *C. glabrata* involved in ROS production, a BLAST search of the protein sequences of *C. glabrata* was carried out by comparison with NADPH-oxidases in *S. cerevisiae* and *C. albicans*. According to the BLAST result, we identified twelve candidate genes as members of the NOX enzyme family in *C. glabrata*. Of these candidates, the protein encoded by *CAGL0K05863g* (*CgNOX1*) showed the highest protein identity (40.94%) to *Sc*Yno1p/*Sc*Aim14p (Appendix A).

In addition, we evaluated the transcript levels of these candidate genes during cultivation. Of the twelve candidates, transcriptional expression of *CgNOX1* and four other candidates was observed (Appendix A). Hence, we selected *CgNOX1* as a putative member of the NADPH oxidase family in *C. glabrata*. To gain further insights into the roles of *CgNOX1* in *C. glabrata*, we conducted this study using the *nox1* null mutant (*Cgnox1∆*) and the *NOX1*-revertant strain.

### 3.2. Role of CgNOX1 in C. glabrata ROS Production

ROS production by the *Cgnox1∆* strain was compared with that of the wild-type strain and the *CgNOX1* reintegrated strain during exponential growth in SC medium. In the CM-H2DCFDA reaction with the ROS level of the three strains, we observed that nearly all cells of the wild-type strain and the *CgNOX1*-reintegrated strain exhibited green fluorescence. In contrast, the *Cgnox1∆* mutant showed significantly lower intensity compared to the other two strains, indicating notably reduced intracellular ROS levels (Figure 1A,B). When the chemiluminescence probe lucigenin was used to detect extracellular ROS, the luminescence signal produced in the *Cgnox1∆* sample was almost half that of the wild-type strain, while the reintegrated *CgNOX1* strain reverted to 80% of the wild-type level (Figure 1C). These results indicated that *CgNOX1* is a major enzyme for both intracellular and extracellular ROS production in *C. glabrata*, although *Cgnox1∆* is still able to produce ROS.

### 3.3. Role of CgNOX1 in the Oxidative Stress Response of C. glabrata

Considering its contribution to fungal ROS production, we hypothesized that *CgNOX1* deletion could also affect the sensitivity of *C. glabrata* to external oxidative stress. The wild-type, *Cgnox1∆,* and *CgNOX1* reintegrated strains were cultured to the exponential phase in SC medium and spotted on SC plates containing different oxidizing agents. There were no growth differences between the wild-type and the *CgNOX1* reintegrated strain. Interestingly, the *Cgnox1∆* mutant exhibited hypersensitivity to menadione and especially H_2_O_2_ compared to the other two strains (Figure 2A). Deletion of *CgNOX1* showed little change in the sensitivity to CHP and diamide.

Furthermore, to gain a better understanding of the function of *CgNOX1* in the oxidative stress response, we investigated the mRNA expression level of *CgNOX1* before and after treatment with H_2_O_2_ and menadione using qRT-PCR. The results showed that the *CgNOX1* gene was strongly induced after H_2_O_2_ treatment, but there was little increase in mRNA level after menadione treatment in the wild-type (Figure 2B). In addition, the level of ROS production with H_2_O_2_ treatment was measured to further explore the nature of *CgNOX1* in response to oxidative stress. Fluorescence assay suggested that the wild-type and *CgNOX1* reintegrated strains exhibited higher levels of ROS under H_2_O_2_ stress (Figure 2C,D). Together, these findings indicated that *CgNOX1* plays an important role in, at a minimum, the response to H_2_O_2_ stress.

### 3.4. Role of CgNOX1 in C. glabrata Ferric Reductase Activity

Nox enzymes are considered to have closely related sequences with ferric reductases (Fre). To test whether *CgNOX1* contributes to ferric reductase activity, we conducted the TTC reduction assay (Figure 3A). The positive control, a *C. albicans* wild-type strain (SC5314 strain), displayed red coloration in the TTC assay, indicating surface ferric reductase activity. However, for the *C. glabrata* strains (wild-type, *Cgnox1∆,* and *CgNOX1* reintegrated strains), no detectable surface ferric reductase activity was observed. The absence of surface ferric reductases in *C. glabrata* corresponds with the results reported by Gerwien et al. [21]. Thus, the quantitative ferric reductase assay in culture supernatants was performed. All three *C. glabrata* strains were cultured in iron-restricted medium and reacted with Fe(III) in the reductase buffer. The reductant DTT was used as a positive control, and the ferric reduction activities of all three strains were detected. However, the *CgNOX1* deletion strain showed a significantly lower ability to convert iron than the wild-type strain by spectrophotometric detection (Figure 3B). This result indicated that the absence of *CgNOX1* led to a decrease in extracellular ferric reductase activity.

### 3.5. Role of CgNOX1 in C. glabrata Biofilm Formation

Since the ability to form biofilms is an important virulence factor for *Candida* species, the biofilms formed by the wild-type strain, the *Cgnox1∆* strain, and the *CgNOX1* reintegrated strain were compared. In the XTT assay, the biofilm formation of the *Cgnox1∆* mutant showed almost the same metabolic activity as those of the wild-type and the reintegrated *CgNOX1* strain (Figure 4A). SEM imaging also showed that the biofilm structures of these three strains were similar. All of the strains could form extensive biofilms with several cell layers (Figure 4B). Therefore, *CgNOX1* is not required for in vitro biofilm formation in *C. glabrata*.

### 3.6. Role of CgNOX1 in Co-Incubation with Human Hepatocytes

Previously, we found that *C. glabrata* as well as *C. albicans* induced human transglutaminase 2 (TG2) activity when co-cultivated with human hepatic cells, while such activity was not induced by the *Cgnox1∆* strain [14]. However, at that time, we had not constructed the reintegrated *CgNOX1* strain and could not confirm whether the gene deletion was the main cause of the induced TG2 activity in hepatocytes. Here, to investigate the expression of *Nox* candidate genes in *C. glabrata* during co-culture with hepatocytes, the transcriptional expression of these twelve genes in wild-type *C. glabrata* was analyzed. Among the twelve genes, only two (*CAGL0C03333g* and *CgNOX1*) were induced; *CgNOX1* was induced at a greater level, and its value was more than four times higher than that in the single-cultured group (Appendix A). To further analyze the time-course changes of *CgNOX1* expression in the wild-type during co-incubation with HC cells, we conducted qPCR to measure the mRNA levels of *CgNOX1* at different incubation times during co-cultivation with hepatic cells. Results showed that the mRNA expression level of *CgNOX1* in the co-cultured group was induced at 12 h and peaked at 24 h. Conversely, the transcript level of *CgNOX1* in the single-cultured group did not exhibit significant changes (Figure 5A). During the 24 h co-incubation, this gene was strongly induced compared to the 4 h single-cultured group.

To confirm the role of *CgNOX1* in the induction of cellular TG activity in human hepatocytes in detail, HC cells were co-incubated with the wild-type, the *Cgnox1∆* mutant, and the *CgNOX1* reintegrated strain. While the *Cgnox1∆* mutant lost the ability to induce cellular TG activity, in this work, it was found that the *CgNOX1* reintegrated strain restored the induction in HC cells. When using a ROS inhibitor, NAC, to compare among the *C. glabrata* wild-type, *Cgnox1∆* mutant, and *CgNOX1* reintegrated strains, the induction of TG activity was blocked (Figure 5B).

As for ROS generation in hepatocytes, co-incubation with both the wild-type and the *CgNOX1* reintegrated strain exhibited significantly higher levels of fluorescence. However, this phenomenon was not observed in the co-incubation with the *Cgnox1∆* mutant. After NAC treatment, all three *C. glabrata* strains could not induce the observed levels of cellular ROS production during co-incubation with HC cells (Figure 5C). This result indicated that *CgNOX1* is associated with ROS production at the HC-fungal interface.

## 4. Discussion

Although previous studies have mainly focused on Nox enzymes in multicellular organisms, some researchers have provided cases in which some unicellular organisms also contain Nox activity and some multicellular fungi lack *NOX* genes [10,11,22]. This is the first study on the *NOX* gene in the pathogenic unicellular *C. glabrata*. *CAGL0K05863g* (*CgNOX1*) in *C. glabrata* is a homolog of *YNO1* in *S. cerevisiae* and shares almost 41% protein identity. *S. cerevisiae* Yno1p has been reported to generate intracellular superoxide [6]. The first identified Nox member, Fre8, of *C. albicans* was recently reported to produce a burst of extracellular ROS in the filamentous form. The Fre8-superoxide contributed to extracellular SOD induction during morphogenesis [11]. In our study, both intracellular and extracellular ROS were detected by CM-H2DCFDA and lucigenin in *C. glabrata* wild-type and mutants. CM-H2DCFDA is a cell-permeant indicator and is oxidized by intracellular ROS to H2DCF, which emits a green fluorescence [23]. Lucigenin is a chemiluminescence probe that cannot cross the cell membrane and is specific for superoxide detection by light emission [24]. The decrease of green fluorescence and luminescence in *Cgnox1∆* mutant cells indicated that this gene contributes to the production of both intracellular and extracellular ROS in *C. glabrata*. However, the remaining appearance of ROS in the mutant suggests that there are other sources of ROS. There are other Nox enzyme candidates in *C. glabrata*, and *CAGL0C03333g* was especially induced during HC cell co-culture. Moreover, the mitochondria in both *S. cerevisiae* and *C. albicans* have been reported to generate ROS that is released extracellularly [25,26]. The process of mitochondrial respiration and even other Nox members in *C. glabrata* may contribute to ROS production.

The *CgNOX1* deletion resulted in lower resistance to several oxidizing reagents, especially H_2_O_2_, compared to the *C. glabrata* wild-type and reintegrated strains, and is distinct from the phenomenon in the *S. cerevisiae YNO1* mutant [10]. Further, *CgNOX1* expression and ROS levels were strongly induced after H_2_O_2_ treatment in the wild-type strain, indicating that *CgNOX1* plays a crucial role in the H_2_O_2_ stress response. Our mRNA expression result is similar to the *noxA* gene expression pattern in *Verticillium dahlia* [27]. In that study, three important downstream genes involved in the oxidative stress response (OSR), catalase (*cat1*), superoxide dismutase (*sod1*), and glutathione reductase (*glr1*), were analyzed under H_2_O_2_ stress in the *∆noxA* mutant. It was observed that all three genes were downregulated by oxidative stress in the mutant, indicating that *V. dahliae* NoxA is implicated in the transcriptional regulation of OSR. Therefore, it would be interesting to explore the relationship between the *NOX* gene and other genes activated in the oxidative stress signaling pathways of *C. glabrata*.

In the fungal kingdom, Nox enzymes are highly similar to ferric reductases (Fre enzymes), and their functions are difficult to predict by sequence analysis alone [28]. Moreover, a close relationship between cellular iron homeostasis and oxidative stress is well reported [29,30]. Although Yno1p/Aim14p in *S. cerevisiae* has similarities to the iron/copper reductase *Sc*Fre8p, it does not show ferric reductase function [10]. As discussed above, another recent paper mentioned *CAGL0K05863g* (*CgNOX1*) in *C. glabrata* [21]. However, this previous report mainly focused on another two *FRE* genes in *C. glabrata* and reported that these two proteins do not confer ferric reductase activity in vitro, which is distinct from the roles of Fre superfamily members in *S. cerevisiae*. These two genes, *CAGL0C03333g* and *CAGL0M07942g*, are homologs of *FRE6* and *FRE8* in *S. cerevisiae*, respectively. The *S. cerevisiae* mutant lacking *FRE8* has been reported to be deficient in growth under low iron and respiration conditions [31]. The transcriptional response of *FRE6* was dependent on the iron levels in the culture medium [28]. For our ferric reduction experiment, the amount of Fe(III) converted by the deletion mutant was significantly lower than that of the wild-type. It is suggested that *CgNOX1* might also contribute to ferric reductase activity in *C. glabrata*, rather than *CAGL0C03333g* and *CAGL0M07942g*.

Transglutaminases (TG) are a group of enzymes that catalyze post-translational protein modifications by forming isopeptide bonds and are responsible for regulating various activities such as cell growth, differentiation, metastasis, and apoptosis [32,33]. In ethanol-treated hepatocytes, significant nuclear TG activity was induced, which led to reduced expression of hepatocyte growth factor receptors and subsequent apoptosis in HC cells [34]. Increased intracellular ROS generation has been reported to induce TG activation in many different types of cells, such as human umbilical vein endothelial cells and Swiss 3T3 cells [35,36]. Our previous study demonstrated that co-incubation of hepatic cells with the opportunistic fungi *C. albicans* and *C. glabrata*, but not non-pathogenic *S. cerevisiae*, led to hepatic cell death by enhanced TG2 activity and high ROS generation [14]. Another study also revealed that *C. albicans* infection causes epithelial cell death, and ROS accumulation is mainly responsible for it [37]. We have provided several lines of evidence demonstrating that this phenomenon was not only related to the source of hepatic ROS but was also mediated by the release of fungal ROS during the host-fungal interaction. The addition of ROS scavengers and the deletion of *CgNOX1* blocked the induction of hepatic TG activity. By constructing the reintegrated strain of *CgNOX1*, the present study revealed that the complement of *CgNOX1* restored ROS generation, inducing hepatic TG2 activation during co-culture of HC cells and *C. glabrata*. A similar phenomenon has been observed in the mutants of *C. albicans FRE8*/*CFL11* [12]. According to RNA-seq analysis of the *C. albicans* transcriptome during mouse kidney infection, the expression levels of most genes encoding Fre enzymes were highly upregulated [38]. Nevertheless, in a transcriptome study of *C. glabrata* RNA by the same research group, the upregulation of related genes was not reported. [39]. Our real-time PCR data showed that the expression level of *CgNOX1* significantly increased in HC cell co-culture compared with a single fungal culture. These results indicated that fungal Nox-derived ROS may contribute to interactions with host cells.

The biofilms of *Candida* species are known to be a crucial pathogenic factor in human infections, are highly resistant to antifungal drugs and host immune responses, and are difficult to eradicate [40]. In this study, we observed that lack of *CgNOX1* had no effect on the metabolic activity and microstructure of in vitro *C. glabrata* biofilms, which is consistent with the result of *FRE8* in *C. albicans* [11]. Our results also suggested that biofilm formation is not a direct factor in the induction of hepatic TG2 activity during co-incubation with hepatic cells. Considering that a *C. albicans FRE8* mutant showed attenuated biofilm formation with few elongated hyphae in a rat catheter model and increased death killed by neutrophils [11], *FRE8*-derived ROS was also proposed as the candidate source of ROS at the host-pathogen interface.

In the above context, the identified *NOX* gene in *C. glabrata* plays important roles in fungal ROS generation, the response to oxidative stress, and ferric reductase activity. It also contributes to the induction of host TG2 activity during co-incubation with human hepatic cells and the fungus, which may be mediated by the production of fungal ROS.

## Figures and Tables

**Figure 1 jof-10-00016-f001:**
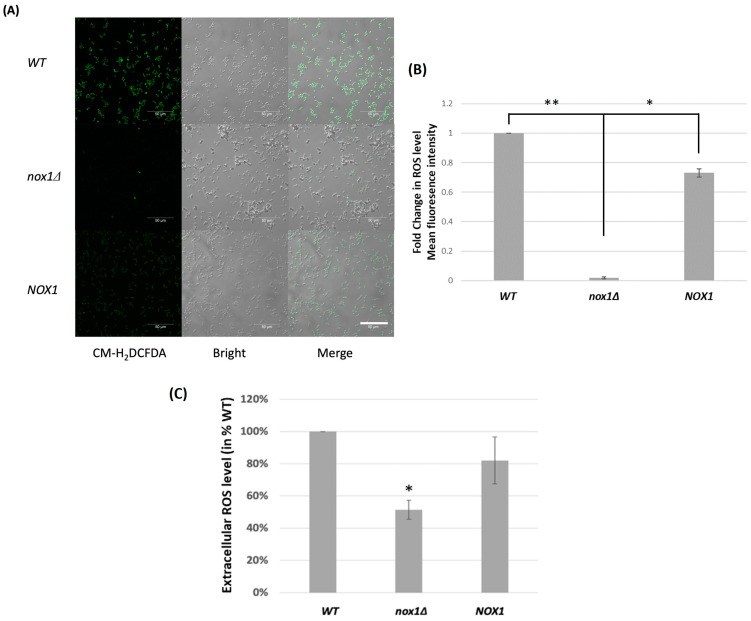
Fungal ROS production in *C. glabrata* wild-type, *Cgnox1Δ* mutant, and *CgNOX1* reintegrated strains Fungal cells were grown to the exponential phase in SC medium. (**A**) Intercellular ROS was reacted with 10 μM CM-H2DCFDA. Green fluorescence signals were visualized by bright-field and epifluorescence microscopy. Scale bars represent 50 μm. (**B**) The mean fluorescence intensities of the indicated strains were quantitated using Image J software (1.53t version). * *p* < 0.05; ** *p* < 0.01 compared with the *Cgnox1∆* mutant. (**C**) Extracellular ROS was detected by 0.1 mM lucigenin, and luminescence was measured using a microtiter plate reader. Data are shown for lucigenin activity at 37 °C in triplicates, expressed as a percentage of the *C. glabrata* wild-type strain. * *p* < 0.05 compared with the wild-type strain and reintegrated strain.

**Figure 2 jof-10-00016-f002:**
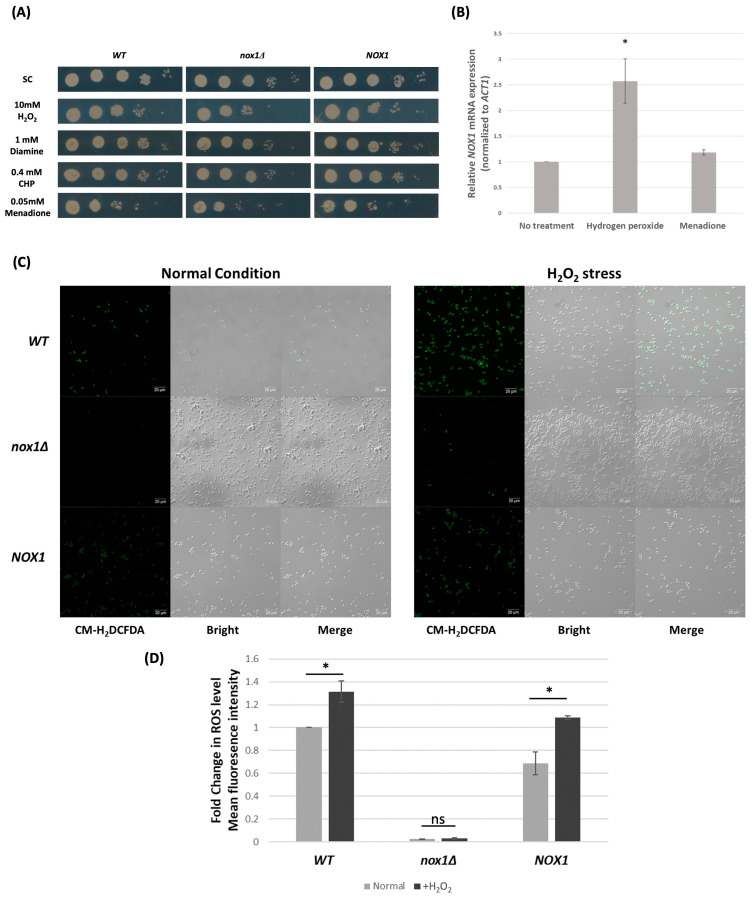
Oxidative stress responses in *C. glabrata* wild-type, *Cgnox1Δ* mutant, and *CgNOX1* reintegrated strains (**A**) Fungal cells were cultured to the exponential phase in SC medium, and serial 10-fold dilutions were spotted on SC plates containing 10 mM H_2_O_2_, 1 mM diamide, 0.4 mM CHP, and 0.05 mM menadione. (**B**) Transcript levels of *CgNOX1* were assessed under three conditions: 10 mM H_2_O_2_ treatment, 0.5 mM menadione treatment, and without any treatment. The *ACT1* transcript level in *C. glabrata* was used for normalization. Reported values indicate the mean ± SD of at least three independent experiments. * *p* < 0.05 compared with the untreated wild-type strain. (**C**) Mid-exponential phase cells were treated with or without 10 mM H_2_O_2_ for 60 min and stained with 10 μM CM-H2DCFDA for 30 min. The fluorescence signal, an indicator of ROS generation, was measured by a Zeiss fluorescence microscope. Scale bars represent 20 μm. (**D**) The mean fluorescence intensities of the indicated strains and conditions were quantitated using Image J software (1.53t version). * *p* < 0.05 compared with normal conditions, and “ns” indicates not significant (*p* > 0.05).

**Figure 3 jof-10-00016-f003:**
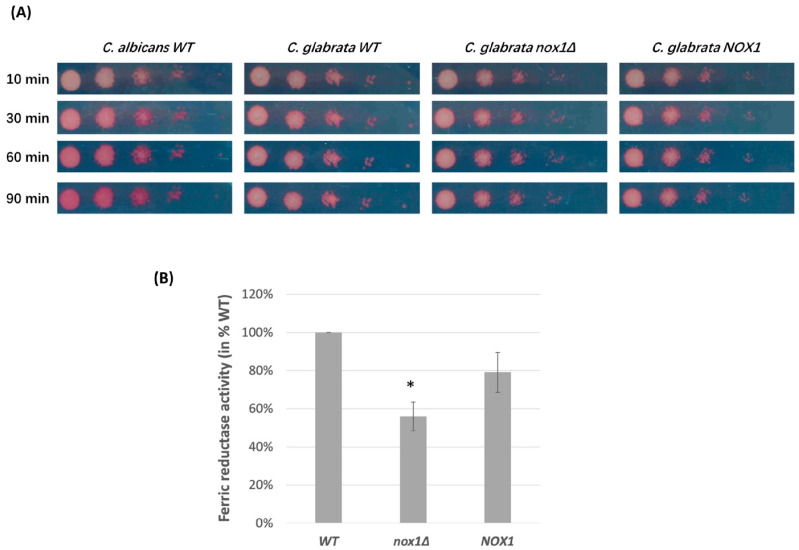
Ferric reductase activities of *C. glabrata* wild-type, *Cgnox1Δ* mutant, and *CgNOX1* reintegrated strains. (**A**) Colony spots of different strains were overlaid with agarose containing 0.1% TTC and incubated at 30 °C for the indicated times before being photographed. (**B**) Fungal cells were cultured to the exponential phase in SC iron-restricted medium and reacted with the Fe(III)-containing reductase buffer. The absorbance was read at 520 nm. Data are shown for extracellular ferric reduction activity in triplicates, expressed as a percentage of the *C. glabrata* wild-type strain. * *p* < 0.05 compared with the wild-type strain and reintegrated strain.

**Figure 4 jof-10-00016-f004:**
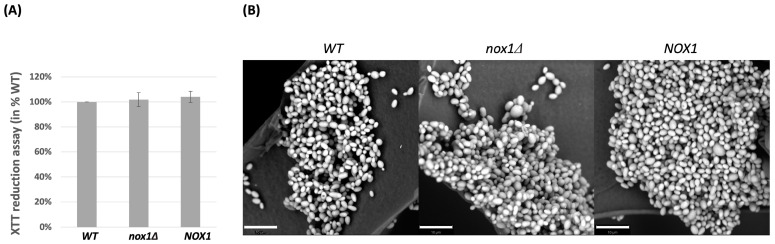
Biofilm formation in *C. glabrata* wild-type, *Cgnox1Δ* mutant, and *CgNOX1* reintegrated strains. Biofilms were formed by the *C. glabrata* strains in 96-well plates for 24 h. (**A**) The metabolic activities of the biofilms of the three strains were detected by the XTT reduction assay at 490 nm. (**B**) The biofilm microstructures of the three strains formed on silicone sponges were imaged by scanning electron microscopy. Scale bars represent 10 μm.

**Figure 5 jof-10-00016-f005:**
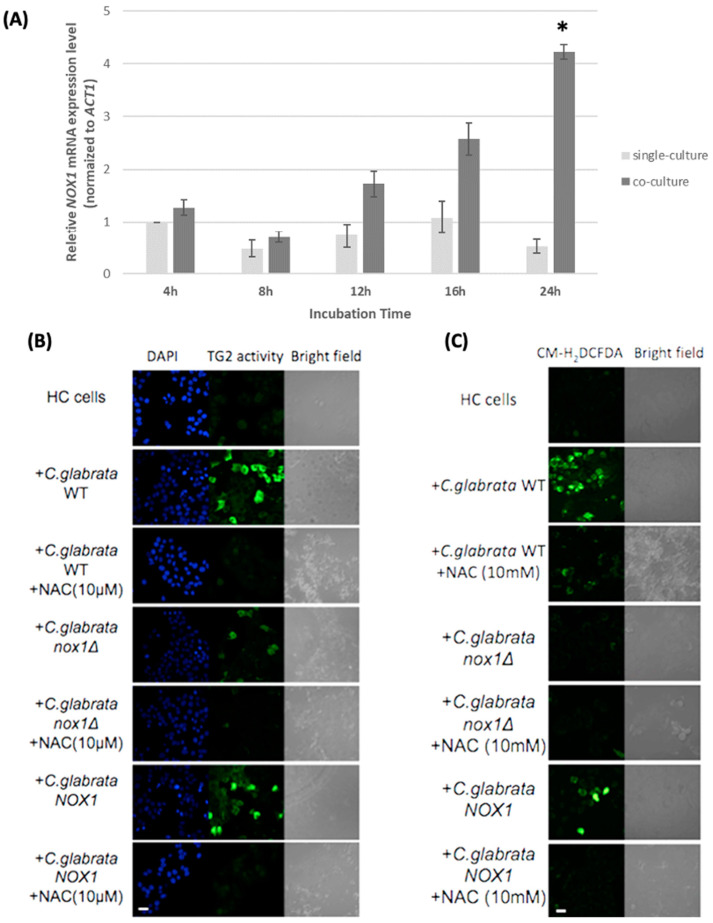
*CgNOX1* enhanced HC cellular TG activity and ROS production, and its expression was significantly upregulated during co-incubation. HC cells (2 × 10^5^ cells/well) were incubated alone or with *C. glabrata* strains (2 × 10^7^ cells/well). In some cases, the fungal cells were treated with 10 mM NAC, a ROS inhibitor. (**A**) Transcript levels of *CgNOX1* were assessed for the specified incubation times under conditions with or without co-incubation with HC cells. The *ACT1* transcript level in *C. glabrata* was used for normalization. Reported values indicate the mean ± SD of at least three independent experiments. * *p* < 0.05 compared with the 4 h single-cultured group. (**B**) TG activity was detected as a green fluorescence signal from TRITC staining 5-BAPA. (**C**) ROS production was analyzed with 5 μM CM-H2DCFDA. The fluorescence signals were visualized by bright-field and epifluorescence microscopy. Scale bars represent 20 μm.

**Table 1 jof-10-00016-t001:** List of *C. glabrata* strains.

Name	Parent	Genotype	Source or Reference
CBS138		Wild type strain	[13]
2000H	2001U	*Δhis3::ScURA3Δura3*	[13]
KUE100	2000H	*his3 yku80::SAT1 flipper*	[13]
*nox1Δ*	KUE100	*his3 yku80::FRT nox1::CgHIS3*	[14]
NOX1	*nox1Δ*	*His3 yku80::FRT NOX1::CgHIS3*	This study

**Table 2 jof-10-00016-t002:** Primers for the construction of plasmids.

Primer Name	Sequence 5′-3′
pHIScheckF	AGAAAACCAGCCTCACGATG
pHIScheckR	GTTCTTCTAGGGGAGCTAGTAGGGG
pNox1compF	GCTCTAGATAGGACTAGATGTAATTGAGCC
pNox1compR	GCTCTAGAGTTGAAGCATTCGGTATTAAC
pChr606 F1	AAGAATGCCAACCAAGGATTCACAATAATCCGAAGC
pChr606 R1	TTAGGCAAAGCATTTGTAAACCATTACAAGCACTC
pZeoORFcheckF1	AAGTTGACCAGTGCCGTTCCGGTG
pchrF606kcheck	CTAATGGGGATATAGAAAGATAGGG

## Data Availability

Data are contained within the article and Appendix A.

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
