# Peer review of "A Putative NADPH Oxidase Gene in Unicellular Pathogenic Candida glabrata Is Required for Fungal ROS Production and Oxidative Stress Response"

_jof, 2023, doi:10.3390/jof10010016_

Round 1
Reviewer 1 Report
Comments and Suggestions for Authors
This paper presents an examination of the C. glabrata strain lacking the ortholog of S. cerevisiae AIM14/YNO1 gene, which encodes a NADPH oxidase-like protein. The gene, which the authors call NOX1, was deleted, and the phenotype of the null mutant was compared to its parental strain, as well as the deletion strain where the gene was inserted elsewhere in the genome (the complemented strain). The mutant was analyzed in several different assays, and it was concluded that it has a deficiency in ROS production under unperturbed conditions, is more sensitive than the WT strain to hydrogen peroxide, and affects C. glabrata ferric reductase activity and transglutaminase (TG) activity of hepatocytes co-incubated with C. glabrata. It was also shown that the NOX1 gene is induced in C. glabrata in response to hydrogen peroxide and co-incubation with hepatocytes. Overall the question of the role of NOX homologs in C. glabrata is interesting, and the approaches to study it are sound, but there are a few issues with the results.
Major:
1. The complemented strain does not appear to fully complement in most of the assays shown. This could be because something is wrong with the re-integrated gene or, more troublingly, the deletion mutant had acquired additional mutations contributing to the observed phenotypes. In the latter case, this would make most of the results inconclusive. This needs to be resolved before the paper can be published. Either additional independent deletions have to be generated and shown to have the same phenotypes, or another complemented strain has to be generated (perhaps using a different strategy) and shown to fully complement.
2. The CFDA fluorescence (ROS levels) results are shown in a qualitative, not quantitative way, for unclear reasons. Usually, CFDA fluorescence is analyzed by flow cytometry, which produces quantitative results of relative ROS levels between strains/conditions. Showing the microscopy images does not provide this necessary quantitativeness. For instance, it is clear that the complemented strain has lower fluorescence/ROS levels than the parental strain, but how much lower? Is it 50%? 25%? This is important for reasons described above in point 1. Also, showing the CFDA results in a quantitative way will help to compare the effects of nox1∆ on intracellular vs extracellular ROS formation.
3. Line 331 and Figure 5C: it is not clear whether the detected ROS are produced by the hepatocytes or the fungal cells or both.
Minor:
1. Lines 99-100: "cells were exposed to 10 mM H2O2 for 60 minutes at 30°C. In the untreated condition, the H2O2 was replaced with SC medium. " makes it seem like the cells were in water + 10 mM H2O2 instead of in SC + 10 mM H2O2 for 60 minutes. Which is correct?
2. Line 210: which medium were the cells cultured in prior to RNA prep - YPD or SC?
Author Response
Dear reviewer,
Thank you for your comments concerning our manuscript “A putative NADPH oxidase gene in unicellular pathogenic Candida glabrata is required for fungal ROS production and oxidative stress response”. Those comments are valuable and helpful for revising and improving our paper, providing important guidance for our research. We have carefully studied comments and have made corrections, which we hope will meet with approval. Revised portions in the paper are highlighted. The main corrections in the paper and responds to the reviewer’s comments are as following:
Reviewer’s comments:
Major:
- The complemented strain does not appear to fully complement in most of the assays shown. This could be because something is wrong with the re-integrated gene or, more troublingly, the deletion mutant had acquired additional mutations contributing to the observed phenotypes. In the latter case, this would make most of the results inconclusive. This needs to be resolved before the paper can be published. Either additional independent deletions have to be generated and shown to have the same phenotypes, or another complemented strain has to be generated (perhaps using a different strategy) and shown to fully complement.
Response to Reviewer: Thank you for pointing out the results about the complemented strain. It is a pity that we could not obtain the full complementation with this strain compared to the wild-type strain. We constructed this complemented strain to express NOX1 with its original promotor and terminator regions by using the vector pZeoi_comp606. This method has been used in several published papers (Ueno et al., 2011, Nagi et al., 2013, Chen et al., 2021). We have confirmed the accurate integration at appropriate recombination locus by PCR, same as the confirmation on the accurate deletion in deletion mutant. The phenotypes of the complemented strain in most assays have exhibited recovery to over 70% of wild-type strain and showed a significant difference from those of the deletion mutant, as we pointed out in the revised manuscript. Thank you for the constructive comments again. We will consider constructing the complemented strain using a different strategy with another promoter in the future.
- The CFDA fluorescence (ROS levels) results are shown in a qualitative, not quantitative way, for unclear reasons. Usually, CFDA fluorescence is analyzed by flow cytometry, which produces quantitative results of relative ROS levels between strains/conditions. Showing the microscopy images does not provide this necessary quantitativeness. For instance, it is clear that the complemented strain has lower fluorescence/ROS levels than the parental strain, but how much lower? Is it 50%? 25%? This is important for reasons described above in point 1. Also, showing the CFDA results in a quantitative way will help to compare the effects of nox1∆ on intracellular vs extracellular ROS formation.
Response: Thanks for constructive advice. The reason why we used microscope to visualize CFDA fluorescence is that the microscopy images can directly show the position of intracellular ROS in both single fungal cell culture samples and co-incubation hepatic samples. We agree that it is necessary to show the quantitative results. Therefore, we used Image J software to present relative semi-quantification data for ROS level, in which Ostu is set as the threshold, and the area mean fluorescent intensity data were calculated for at least 50 individual cells. The results showed that the mean fluorescent intensities of nox1∆ mutant were significantly lower than WT and complemented strain in both intracellular and extracellular experiments. The complemented strain recovered more than 70% of the parental strain. We added these results and figures in our manuscript.
- Line 331 and Figure 5C: it is not clear whether the detected ROS are produced by the hepatocytes or the fungal cells or both.
Response: This experiment was detected ROS produced by the hepatocytes and we revised the text of this part.
Minor:
- Lines 99-100: "cells were exposed to 10 mM H2O2 for 60 minutes at 30°C. In the untreated condition, the H2O2 was replaced with SC medium. " makes it seem like the cells were in water + 10 mM H2O2 instead of in SC + 10 mM H2O2 for 60 minutes. Which is correct?
Response: Sorry for the misunderstanding. The treated cells were in SC medium + 10mM H2O2 for 60 minutes. The manuscript has been revised.
- Line 210: which medium were the cells cultured in prior to RNA prep - YPD or SC?
Response: For the Candida under the oxidative stress, the cells were cultured in SC medium before RNA prep. For the cells co-incubated with hepatocytes, the cells were precultured in YPD medium, washed by PBS and co-incubated with hepatocytes in the phenol red free DMEM medium.
We would like to thank you for allowing us to resubmit a revised copy of the manuscript and we highly appreciate your time and consideration.
Best Wishes,
Xinyue Chen
Reviewer 2 Report
Comments and Suggestions for Authors
The authors identify a novel NOX ortholog in Candia glabrata. Intra- and extracellular ROS production and oxidative stress response was compared in wild type strain, delta nox1 mutant strain and retransformed mutant strain with wild type copy of NOX1.
The authors show the influence on ROS production and impaired ferric reductase activity. No effect was found on biofilm formation.
Minor points:
Materials and Methods
Line 74: “supplement mixture (CSM)” The authors should describe the precise formula or the manufacturer of the supplement. The authors should include a reference if they do not develop the supplement mixture.
Line 87: “primers pNox1compF and pNox1compR” Primer sequences missed in Table 2
Line 91: “by the usual lithium acetate method”. I missed a reference for the method.
Line 107: “phosphate-buffered saline (PBS) buffer” The authors should define the formula or include manufacturer information.
Line 125: “in TAE buffer” as mentioned before
Line 195: “relative expression of genes…” I missed the reference for the method.
Figure 1B I miss the description of Y-axis in the figure legend.
Figure 2B
There is a discrepancy between Figure2B and Figure S1, both y-axes show relative expression and were normalized to ACT1 transcript levels. The values for NOX1 are different. In Figure 2B, is the transcript level of 1 correct for Actin as well for NOX1 without treatment?
Author Response
Dear reviewer,
Thank you so much for your comments and professional advice. Those opinions are valuable and helpful for revising and enhancing our paper. Based on your suggestions and requests, we have made corrections in the revised manuscript, highlighted for your convenience. The main modifications and responses to the comments are presented below.
Comments:
Materials and Methods
Line 74: “supplement mixture (CSM)” The authors should describe the precise formula or the manufacturer of the supplement. The authors should include a reference if they do not develop the supplement mixture.
Response: Thank you for your suggestion. The CSM was purchased from funakoshi (Tokyo, Japan), which has been added in the manuscript.
Line 87: “primers pNox1compF and pNox1compR” Primer sequences missed in Table 2
Response: The primers CgNOX1_XbaI_F/CgNOX1_XbaI_R in the previous Table 2 are pNox1compF/ pNox1compR. The Table has been revised.
Line 91: “by the usual lithium acetate method”. I missed a reference for the method.
Response: Thank you for your comment. The reference for the method has been added.
Line 107: “phosphate-buffered saline (PBS) buffer” The authors should define the formula or include manufacturer information.
Response: Thank you for your suggestion. The PBS buffer formula has been added.
Line 125: “in TAE buffer” as mentioned before
Response: The TAE buffer formula has been added.
Line 195: “relative expression of genes…” I missed the reference for the method.
Response: The reference for the method has been added.
Figure 1B I miss the description of Y-axis in the figure legend.
Response: The description of Y-axis in Figure 1B as well as Figure 3B has been revised in figure legend as suggested.
Figure 2B
There is a discrepancy between Figure2B and Figure S1, both y-axes show relative expression and were normalized to ACT1 transcript levels. The values for NOX1 are different. In Figure 2B, is the transcript level of 1 correct for Actin as well for NOX1 without treatment?
Response: Thank you for your criticism. In Figure 2B, the relative expression level was normalized to ACT1 transcript level and compared with the untreated group, where the relative level was set as 1. However, in Figure S1, all transcripts were normalized solely to the ACT1 transcript, with the ACT1 expression level set as 1. We have revised Figure S1 legend.
We would like to thank you for allowing us to resubmit a revised copy of the manuscript and we highly appreciate your time and consideration.
Best Wishes,
Xinyue Chen
Reviewer 3 Report
Comments and Suggestions for Authors The Authors indicate the important role of the NOX gene in Candida glabrata in the response to oxidative stress and iron reductase activity,as well as its participation in the interaction with host cells.
Author Response
Dear Reviewer,
Thank you for taking out of your busy schedule to review our manuscript entitled “A putative NADPH oxidase gene in unicellular pathogenic Candida glabrata is required for fungal ROS production and oxidative stress response”. We will continue to strive for improvement in our future research. Again, we highly appreciate your time and consideration.
Best Wishes,
Xinyue Chen
Round 2
Reviewer 1 Report
Comments and Suggestions for Authors
My concerns have been addressed.